# Characteristics, Preventive Factors, and Barriers to Breastfeeding and Mixed Feeding in the First Month of Life in Barcelona: The Multicenter Observational Study GREEN MOTHER

**DOI:** 10.3390/nu17193109

**Published:** 2025-09-30

**Authors:** Azahara Reyes-Lacalle, Rosa Maria Cabedo-Ferreiro, Judit Cos-Busquets, Liudmila Liutsko, Margalida Colldeforns-Vidal, Rosa García-Sierra, Mª Mercedes Vicente-Hernández, Miriam Gómez-Masvidal, Laura Montero-Pons, Encarnación López-Gimeno, Pere Torán-Monserrat, Gemma Falguera-Puig, Gemma Cazorla-Ortiz

**Affiliations:** 1Research Group on Sexual and Reproductive Healthcare (GRASSIR) (2021-SGR-793), 08204 Sabadell, Spain; areyesl.ics@gencat.cat (A.R.-L.);; 2Reproductive and Sexual Healthcare (ASSIR), Catalan Institute of Health (ICS), 08204 Sabadell, Spain; juditcos.mn.ics@gencat.cat; 3Reproductive and Sexual Healthcare (ASSIR), Catalan Institute of Health (ICS), 08402 Granollers, Spain; 4Metropolitan North Research Support Unit (USR), Jordi Gol i Gorina University Institute Foundation for Primary Health Care Research (IDIAP), 08303 Mataró, Spain; rgarciasi.mn.ics@gencat.cat (R.G.-S.);; 5Multidisciplinary Research Group in Health and Society (GREMSAS) (2021-SGR-0148), 08007 Barcelona, Spain; 6Reproductive and Sexual Healthcare (ASSIR), Badalona—Sant Adrià, Catalan Institute of Health (ICS), 08930 Sant Adrià del Besós, Spain; 7Reproductive and Sexual Healthcare (ASSIR), Mataró, Catalan Institute of Health (ICS), 08302 Mataró, Spain; 8Reproductive and Sexual Healthcare (ASSIR), Dreta, Catalan Institute of Health (ICS), 08009 Barcelona, Spain; 9Faculty of Nursing, Universitat de Barcelona, 08907 L’Hospitalet de Llobregat, Spain; 10The Institute for Health Science Research Germans Trias i Pujol (IGTP), 08916 Badalona, Spain; 11Department of Medicine, Faculty of Medicine, Universitat de Girona, 17003 Girona, Spain

**Keywords:** breastfeeding, mixed milk feeding, formula milk, infant newborn, maternal-fetal medicine, bottle feeding, neonatology, child nutrition sciences, breastfeeding preventive factors

## Abstract

**Background/Objectives:** Breastfeeding is the best source of food for newborns. Leading health organizations recommend exclusive breastfeeding for the first 6 months of life, followed by the gradual introduction of complementary foods. Evidence shows that breastfeeding offers numerous benefits for newborns, mothers, society as a whole, and the environment. Current breastfeeding rates fall below the established recommendations. This study aims to describe breastfeeding rates at hospital discharge and one month postpartum, analyze fluctuations in feeding types during this period, and identify any characteristics, preventive factors, and barriers to breastfeeding and mixed feeding. **Methods:** This is a multicenter observational study in the North Metropolitan area of Barcelona, with 411 participants surveyed by midwives. **Results:** In total, 79% of women were exclusively breastfeeding, 14% practiced mixed breastfeeding, and 7% used formula feeding at hospital discharge. At one month postpartum, these rates fluctuated to 64%, 23%, and 13%, respectively. Factors such as older age, having a university education, having 16 weeks of parental leave, and having a foreign-born status were positively associated with exclusive breastfeeding, while mental illness was associated with a lower prevalence. The use of breastfeeding accessories was not significantly associated with fluctuations in feeding type, but it was associated with maintaining mixed feeding. **Conclusions:** It is essential to implement individual and community interventions, educate healthcare professionals on factors that hinder breastfeeding, and promote workplace policies that support breastfeeding.

## 1. Introduction

The World Health Organization (WHO), Baby Friendly Hospital Initiative (BFHI), United Nations International Children’s Emergency Foundation (*UNICEF*), and Spanish Association of Pediatrics (AEP) recommend exclusive breastfeeding (EBF) until 6 months of age, followed by continued breastfeeding or total breastfeeding (TBF). TBF refers to any type of BF (including EBF and MF), whether provided directly at the breast or through accessories (breast pumps, nipple shields, supplemental nursing systems, bottles, etc.) [1,2], with the gradual introduction of complementary foods, continuing as long as mutually desired by mother and child [3,4]. One of the WHO’s goals for 2025 was to increase the EBF rate to at least 50% during the first 6 months [5].

BF provides enormous benefits to newborns, mothers, society, and the environment. Breastfed newborns present fewer respiratory [6] and digestive infections, a lower rate of malocclusions, and possible cognitive advantages [7], as well as a probable reduction in the medium- and long-term risk of suffering from overweight, diabetes [5,8,9], and cardiovascular diseases [10]. BF protects mothers against breast and ovarian cancer and type 2 diabetes and contributes to natural birth spacing [5,9,11]. As for society in general, BF reduces the probability of hospitalizations in early childhood, which lowers economic and healthcare costs [12]. Lastly, BF is 500 times more sustainable than formula feeding (FF) in terms of carbon footprint, water consumption, and water scarcity [13]. The benefits of BF compared to FF are well documented.

The *UNICEF* report “Breastfeeding: A Mother’s Gift, for Every Child” [3] shows that 95% of babies are breastfed at some point in their lives. Only 4% of babies were never breastfed in low- and middle-income countries, and 37% received EBF at 6 months; whereas BF prevalence was worse in high-income countries: 21% were never breastfed, and 20% continued to receive EBF at 1 year [3,9]. 

In Spain, less than 80% of babies initiated breastfeeding, which represents the lowest rate among high-income countries [9]. In 2017, the Spanish National Statistics Institute (INE) reported the following rates for infant feeding type at 6 weeks after birth: EBF 68%, mixed feeding (MF) 7%, and FF 27% [14]. According to data from Madrid in 2018, the prevalence of BF and EBF at hospital discharge was 88% and 78%, respectively [15]. In the 2019 Lactem study in Catalonia, 95% of pregnant women expressed their desire to BF for at least one year [16]. At the time of hospital discharge, 94.8% of mothers were BF, of whom 75% were EBF. At one month after birth, these rates had fluctuated to 65% practicing EBF, 24% MF, and 12% FF [16].

Mixed feeding (MF) refers to breastfeeding that is supplemented with formula feeding (FF) to meet the nutritional needs of the infant and is often used during the transition period or in response to challenges with EBF. It is a common infant feeding practice around the world; however, there is not enough data about this type of infant feeding to help and orient mothers practicing it and achieve long-term EBF or BF. The global prevalence of MF is estimated to be 23% to 32% during the first year of life [17]. Other studies report rates as high as 50% or more, which seem to increase in upper–middle-income countries [18]. Early BF cessation is a major problem in Western societies, which is why the *UNICEF* UK Guide to the Baby-Friendly Initiative Standards recommends that BF be continued when EBF is not possible, even if only partially [19]. 

A recent systematic review identified the most common reasons mothers opt for MF, including factors related to the perceived need for supplemental formula (39% of cases: perceived infant hunger, insufficient breast milk, or difficulty breastfeeding), perceived choice (34%: greater flexibility), and perceived pressure (25%: returning to work or receiving advice from healthcare professionals) [18].

In Australia, Reynolds reported that 94% of women intended to start breastfeeding; of these, 95% did so, but only 57% were practicing EBF at 6 months. Women with lower education levels, higher BMI, and smoking habits were less likely to continue EBF. They reported that the reason for quitting EBF was due to BF challenges in 47% of cases and low milk production in 40% of cases. This study also analyzed sociodemographic characteristics alongside the reasons for BF cessation [20]. ML Gianni obtained similar results, with 70% of women reporting BF difficulties (cracked nipples, perceived hypogalactia, breast pain, fatigue, and breast engorgement), and 63% of mothers reported these issues within the first month postpartum. Gianni et al. [21] associated the perception of insufficient milk and mastitis with the risk of not continuing EBF at 3 months. In contrast, vaginal delivery and professional and family support were protective factors [21]. Odom also described factors associated with BF cessation: being a single mother, a multiparous woman, or having a low education level. The study concluded that 60% of mothers who quit breastfeeding did so earlier than desired due to BF difficulties (pain, cracks, engorgement, and mastitis), feeling they did not have enough milk, needing to take medication, and the strain of pumping breast milk [22]. In 2014, Brown analyzed BF cessation before 6 weeks in 500 women: 25% cited discomfort and maternal fatigue, and 24% cited insufficient milk supply as reasons for early cessation [23]. In short, both maternal decisions about infant feeding type and its duration are multifactorial.

Based on these data, we aimed to analyze changes in exclusive, mixed, and formula feeding between hospital discharge and one month postpartum and any associated factors in the region of Catalonia. This article is part of a larger project investigating BF, healthy eating, and sustainability: the GREEN MOTHER project [24].

## 2. Materials and Methods

### 2.1. Design and Study Setting

This was a multicenter, prospective observational cohort study with consecutive sampling of pregnant women in their third trimester and postpartum mothers at their first postpartum checkup receiving prenatal care at Reproductive and Sexual Health Care units (ASSIR, from the Spanish Atenció a la Salut Sexual i Reproductiva). The seven ASSIR units of the Barcelona Metropolitan North area participated in this study. These are located in the primary care facilities of the public health system in Catalonia (Spain).

### 2.2. Sampling and Study Population

The study population comprised pregnant women in their third trimester and postpartum women being seen at Reproductive and Sexual Health Care units in primary care. Data were reported by the women recruited and collected by midwives from November 2022 to April 2023.

The number of participants required for this study was calculated using GRANMO software, version 8 (“Calculadora de Tamaño muestral GRANMO,” n.d.; MA Lighting International GmbH; Paderborn, Germany). A sample of 150 participants is sufficient to estimate an expected population percentage of approximately 50%, with a 95% confidence interval and a precision of +/− 5%. From our previous studies [16], we know that with an initial recruitment of 400 women, there might be 256 (64%) practicing EBF, 92 (23%) practicing MF, and 52 (13%) practicing FF at follow-up one month later [24]. The expected number of patients in each type of breastfeeding would allow replacement rates for losses to follow-up of up to 35%.

Of the 429 study participants, we included 411 in the analysis, as 18 participants did not provide information on feeding type at 1 month postpartum.

### 2.3. Inclusion and Exclusion Criteria

The inclusion criteria were voluntary agreement (including signed informed consent) to participate in the study (the option to participate in the study was offered to all pregnant women in their third trimester and postpartum mothers at their checkup at 7 to 14 days postpartum) and being at least 16 years old. The exclusion criteria were language barriers (not speaking Catalan or Spanish) that impeded informed consent and data collection. However, illiterate women who could communicate verbally in these languages were included, with assistance from the principal investigator in completing the questionnaires.

### 2.4. Procedures

The principal investigator trained the recruiting midwives on how to complete the surveys and fill out the study variables on the Research Electronic Data Capture (REDcap) platform, a secure online application [25]. Some of these data were already available in the Catalan electronic clinical record program (eCAP, Estació Clínica d’Atenció Primaria). Other data were obtained directly from participants who self-completed the questionnaires through the REDCap platform with the help of midwives. All participants gave written informed consent after receiving complete information about the study from the midwives. The ICS (Catalan Health Institute) is responsible for keeping the data secure.

### 2.5. Measurements

Three structured questionnaires were designed to collect descriptive data during three visits: recruitment, first postpartum visit (1st week), and late postpartum visit (1 month) (more details are available in the Protocol phase 1 [24]):

1. Maternal sociodemographic characteristics;

2. Clinical data on the birth;

3. Infant feeding practices, including the use of feeding-related accessories.

### 2.6. Data Analysis

A univariate analysis was used for descriptive statistics. The data were presented for categorical variables as frequencies (n, %) and for continuous variables as ranges, means (SD), or medians (95% CI) for variables with normal and non-normal distribution. Normality distribution for continuous variables was checked by the Shapiro–Wilk test. Dynamic changes in infant feeding types between hospital discharge and 1 month postpartum were quantified by differences in their respective frequencies. A bivariate (chi-square) analysis was performed to detect the most relevant sociodemographic and clinical factors related to feeding type at 4–6 weeks postpartum. Lastly, a multivariate analysis (mlogit) with risk relative ratios (RRRs) was performed to observe the magnitude of associations between the study factors and 4–6 weeks postpartum feeding types. The data analysis was performed with STATA v.18.

### 2.7. Ethics

This project was approved by the Research Ethics Committee of the Jordi Gol i Gorina University Institute Foundation for Primary Health Care Research (IDIAP) under code 22/101-P dated 22/02/2023. This study follows the guidelines of the Declaration of Helsinki. The data were collected anonymously through questionnaires on the REDCap platform. The variables needed to conduct the study were obtained directly from the project participants with their consent, in accordance with the provisions of articles 6.1.a) and 9.2.a) of the GDPR. Anonymous data collection ensured compliance with Regulation (EU) 2016/679. The ICS is the data controller and the current study’s data owner.

## 3. Results

### 3.1. Sociodemographic and Clinical Description of the Participants

The main sociodemographic and clinical data for study participants are presented in Table 1. The mean age of study participants was 33.2 (±5.2) years. A total of 228 (56%) participants were between 26 and 36 years old. In terms of educational level, 149 (44%) participants had a university education, while 4% had no education. A total of 398 (97%) participants had a partner. Regarding place of birth, 291 (71%) participants were born in Spain. A total of 209 (54%) participants had no previous children. As for employment activity, 343 (84%) participants had paid maternity leave. This leave lasted for 16 weeks (the standard duration in Spain) or more for 331 (98%) of them. Regarding paternity leave, 331 (82%) had some time, and of these, 90% had 16 weeks or more. A total of 113 (28%) participants reported having a pathology, and 20 (5%) reported having more than one pathology. As for the infants, 368 (91%) had a normal birth weight (2500–3999 g), and 389 (96%) were born at term (37–42 weeks of gestation). A total of 60 (15%) had a neonatal pathology, and of these, 15 (25%) had more than one pathology.

### 3.2. Infant Feeding Results

#### 3.2.1. Prevalence, Fluctuations, and Characteristics of Infant Feeding in the First Month of Life

The prevalence of infant feeding types at hospital discharge and one month postpartum is shown in Figure 1.

Figure 1 shows a decline in exclusive breastfeeding (EBF) from 79% at hospital discharge to 64% at one month postpartum (χ^2^ = 281.8, *p* < 0.001), while mixed feeding (MF) increased from 14% to 23%, and formula feeding (FF) rose from 7% to 13%.

The feeding trajectory changes in infant feeding type from hospital discharge to one month postpartum were analyzed. Of the mothers practicing EBF at discharge, 59 (14%) had switched to MF and 17 (4%) to FF at one month. Of those practicing MF at discharge, 14 (3%) had switched to EBF and 7 (2%) to FF one month postpartum (Table 2).

#### 3.2.2. Characteristics of the Different Types of Infant Feeding

Regarding the volumes of formula administered at one month of age, MF newborns received a median (95% CI) of 270 mL (180–329 mL) (Table 3), and FF newborns received 845 mL (770–904 mL) (Table 4).

Table 3 shows the distribution of formula volumes administered to MF newborns at one month of age, including the number of daily feedings with their respective volumes. In MF babies, 46% took 70 mL or less at each feeding, 32% took 90 mL supplements, and 22% had more than 90 mL. In total, 61% of babies were fed formula one to four times per day, with the most frequent being a single feeding (20%), while 45% of infants were fed formula three times per day or less (Table 3).

Table 4 shows the feedings of FF newborns at one month of age. In FF, 98% of feed volumes were 90 mL or more, and 83% of these ranged from 90 to 130 mL. In total, 87% of babies were fed seven to nine times a day.

#### 3.2.3. Breastfeeding Accessories

A total of 153 (37%) participants used breast pumps. In total, 94 (35%) exclusively breastfeeding mothers used a breast pump, as did 55 (59%) of those practicing MF. Of the 72 mothers who used nipple shields (18% of total participants), 41 were EBF (57%), 29 were MF (40%), and 2 were LA (3%). A total of 13 (6%) mother–child dyads used a feeding tube or supplemental nursing systems to administer milk supplements, 7 of whom were EBF, 5 were MF, and 1 was LA.

A bivariate analysis was performed to determine the relationship between the use of accessories and fluctuations in infant feeding type. We observed significant differences (*p* < 0.001), and the use of a breast pump was associated with maintaining the infant feeding type rather than changing it. In women who maintained MF throughout the observation period, 73% used a breast pump.

The use of nipple shields was not significantly associated with cessation or improvement in infant feeding type, but there was a relationship between the use of nipple shields (*p* = 0.005). There was no statistically significant association between the use of feeding tubes or supplemental nursing systems and changes in infant feeding type (*p* = 0.329).

### 3.3. Association Between Sociodemographic and Clinical Factors and Infant Feeding Type at One Month Postpartum

The results of the bivariate analysis of sociodemographic and clinical data and infant feeding type at one month postpartum showed which factors were significant. Older women (*p* = 0.015) and those with a partner who had paternity leave (*p* = 0.048) were associated with a higher prevalence of EBF. The bivariate analysis also showed significant differences in the distribution of infant feeding type by place of origin (Spain vs. outside of Spain), with higher EBF and MF and lower FF in participants born in other countries (*p* = 0.001). The existence of maternal illnesses (hypertension, mental illness, or breast surgery) was associated with a lower prevalence of EBF (*p* < 0.05) (Table 5). Mothers under 25 years old had a lower rate of EBF (44% vs. 65% in other age groups) and practiced FF almost three times more (32% vs. 11–12% in other age groups).

### 3.4. Multivariate Analysis of Sociodemographic and Clinical Factors and Infant Feeding Type at One Month Postpartum

To determine which factors were significant for each feeding type (Table 5), we performed a multivariate analysis on the feeding types at one month. Several factors were significantly associated with BF at one month compared with FF as the baseline reference category using relative risk ratios (RRRs). Positive associations with EBF and MF were found in the following cases:Place of birth outside Spain showed a strong direct positive association with a higher probability of EBF (RRR = 14.4; 95% CI: 3.5–58.4) and MF (RRR = 17.2; 95% CI: 4.0–73.2).Older maternal age was associated positively with higher odds of both EBF (RRR = 1.8; 95% CI: 1.01–3.30) and MF (RRR = 2.2; 95% CI: 1.1–4.2).Availability of parental leave was significantly positively associated only with higher probability of EBF (RRR = 3.3; 95% CI: 1.3–8.3).

Regarding maternal conditions, the following negative associations were identified:Diagnosed maternal mental illness was significantly associated with a lower likelihood of EBF (RRR = 0.12; 95% CI: 0.03–0.58).Other pathologies that showed marginally significant associations with a lower likelihood of EBF were breast surgery (RRR = 0.09; 95% CI: 0.01–1.08) and maternal hypertension (RRR = 0.29; 95% CI: 0.07–1.22), with the latter having a higher probability.

In summary, the multivariate analysis revealed that sociodemographic factors such as maternal age, place of birth, and parental leave availability were positively associated with exclusive and mixed breastfeeding, while certain maternal health conditions were linked to lower breastfeeding rates. These findings highlight the multifaceted nature of infant feeding decisions and set the stage for a deeper exploration of their implications in the following discussion.

## 4. Discussion

This study analyzes the prevalence of different types of infant feeding and the fluctuations between them from birth to one month of age. It also identifies the sociodemographic, clinical, and accessory use characteristics associated with the feeding types. The results for infant feeding rates during the first month of life (EBF 64%, MX 23%, and FF 13%) seem to follow the trends of high-income countries, with a positive attitude towards BF. However, our findings also indicate a significant early postpartum decline in EBF, with a noticeable shift toward mixed and formula feeding in the first month. This trend may reflect early BF difficulties or return-to-work pressures. Compared to the WHO goal of maintaining EBF rates at 50% up to six months postpartum, the current rates remain above this target. However, our data refer to the first month postpartum, and there are still five months remaining, which may influence the overall trend.

Young mothers under 25 years of age comprised 8% of the study population and were less likely to practice EBF (44% vs. 66% in the older group); young mothers were more likely to practice FF. In a study in Brazil, Ferreira associated EBF with young age (20–30 years old), having a partner, having a good level of education, and having multiple pregnancies. However, in the discussion, they described a greater risk of early weaning in young mothers under 20 years of age. Liu, in China, corroborated this [26,27]. Thus, young maternal age could be a risk factor for early weaning or less EBF.

Participants with a high level of education practice EBF at a higher rate (74% vs. 54%), suggesting that they have greater health literacy. It is conceivable that the higher the level of education, the deeper the understanding of the benefits of BF for maternal and child health, and thus the more informed the decision-making processes. The trend suggested by the multivariate analysis is that older age and higher education correlate with a greater likelihood of BF. Ramiro González et al. associated being over 35 years of age, having medium–high economic status, being an immigrant with less than 10 years of residence in Spain, and having participated in a breastfeeding workshop after giving birth with increased breastfeeding [15]. This finding was also reported in the LAyDI study in Spain [1].

In 2023, 34.75% of babies were born to women with non-native status [28], much like in our study, in which 29.02% of participants were born outside the country. Native and non-native participants were practicing EBF at a similar rate (62% vs. 65%), whereas FF was four times higher in the population born in Spain. Similarly, a meta-analysis of 29 studies found no association between a non-native status and starting BF but did identify a relationship between a non-native status and a longer duration of any type of BF, even when considering various confounding factors. The authors note that non-native participants’ behavior depends on the receiving country, their previous culture, and how long they have lived in the host country. There may be a relationship between BF, culture, and the economic status of the non-native population [29].

According to some studies, single mothers stop breastfeeding earlier than desired [30,31]. In our study, 3% of women reported not having a partner, and there was no difference in the prevalence of BF among them. Nor was the number of previous children significant in the bivariate analysis. The LAyDI study, however, found it was associated with the start of BF [1].

Sayres et al. reported that the mode of childbirth, maternal socioeconomic status, return to work, and prenatal education are factors that influence BF [32]. In our study, 84% of mothers worked outside the home, and almost all had maternity leave (16 weeks). In total, 81.73% of partners had paternity leave for 16 weeks as well, which significantly influences EBF rates. In 2022, Martinez-Vazquez corroborated that partner support contributes to EBF [33]. The maternity leave status was not associated with the initial type of infant feeding in our study. This may be because a majority of participants (84%) already had paid leave, and mothers who do not have maternity leave usually stay at home in Spain (maternity leave is primarily related to having paid work before delivery). By contrast, Rimes in Brazil and Steurer and Dagher in the US associated having maternity leave with higher EBF rates [31,34,35]. Tomori added that family and community support also promote BF and insisted on the need to increase breastfeeding support policies at the state and workplace levels [36]. Aubel studied the role that men and other family systems play in BF [37].

In total, 3.43% of mothers reported having mental illness, which was decisive in the decision to practice FF, as corroborated in other studies [29,38,39]. Early detection and follow-up of mental illnesses is necessary to increase EBF rates and improve its duration in these mothers. Although only three participants had undergone breast surgery, two were unable to breastfeed. Surgery can reduce milk production due to the removal of part of the mammary gland or undetected preexisting hypoplasia [40]. In our study, high blood pressure was also associated with increased FF. Other authors have associated lower EBF rates with chronic maternal disease [41], and the LAyDY study found that the absence of disease during pregnancy was associated with TBF at 15 days of life [1].

Skaaning D reports that only 13% of premature Danish infants receive EBF [42]. Low birth weight and prematurity have been linked to difficulty in achieving EBF, but we observed no such difference in our study, which could be because the sample (4.4% premature born) lacked enough variation or power [43,44,45].

Feeding type fluctuations during the first month were observed in exclusive breastfeeding (EBF) and mixed feeding (MF), while formula feeding (FF) remained unchanged from birth. A total of 92.70% initiated any type of breastfeeding (TBF), and 83.86% continued at one month, with MF rates nearly doubling. Despite its relevance, MF is rarely analyzed in detail. In our study, EBF declined from 78.59% at birth to 63.75% at one month, while MF increased to 23% and FF to 13%. Similarly, a study in Milan reported a drop in EBF from 95% to 73%, with MF at 20% and FF at 7% [21]. The LAyDI study, which followed a 2017 Spanish cohort until age two, found that 90.7% initiated breastfeeding, with EBF rates of 66.4% at 15 days and 35% at six months [1]. TBF at six months was 62%, with a median duration of 6.0 months (95% CI: 6.0–6.1). A 2019 study in the same context reported similar TBF rates: 94% at birth, 88% at one month, and 63% at six months [16]. In this study, the TBF rate was slightly lower, at 87% at one month of age.

Between BF at discharge and at one month postpartum, the greatest fluctuation was from EBF to MF. In 2021 in Bizkaia, López de Aberasturi obtained an EBF rate of 77% and an MF rate of 13% at discharge; these rates are similar to our study, with their data also coming from Spain in the post-pandemic years [46]. The use of breastfeeding accessories among mothers practicing mixed feeding (MF) was high, with 59% using breast pumps and 31% using nipple shields. These associations were statistically significant in maintaining MF at one month of age. Breastfeeding support accessories have been shown to help increase milk production, reduce pain, and ensure adequate intake by the newborn [1]. However, in the current study, the use of nipple shields, breast pumps, feeding tubes, and supplemental breastfeeding systems was not significantly associated with changes in infant feeding type.

This finding is controversial, since professionals are divided on whether nipple shields should be used. The concern is that they may reduce milk transfer to the baby, impair sucking, and reduce BF duration. But these factors also generate confusion, since nipple shields are often used by mothers experiencing breastfeeding difficulties. Coentro et al. obtained experimental data suggesting that nipple shields reduce the volume of milk extracted with a pump. Both we and they recommend close monitoring and support for nipple shield users due to these hampering effects [47].

This study analyzed the frequency and use of formula supplements in one-month-old infants receiving MF. The median daily volume in MF (270 mL) was one-third of the median in FF (845 mL), and almost 50% of women were offered three supplements a day of less than 90 mL per bottle (Table 3). Therefore, it can be assumed that MF is used as a supplement to BF in the first month when babies do not gain enough weight. In our study, the use of nipple shields, breast pumps, feeding tubes, and supplemental breastfeeding systems did not significantly influence changes in infant feeding type. Moreover, some mothers also commented on using MF to rest or to be away from the baby [18]. The various factors related to maintaining EBF must be analyzed to understand what helps maintain EBF and what hinders it. Some of these factors will be impossible to modify, but others could be addressed to achieve better BF adherence [1].

### Study Limitations

There was a lower proportion of participants with non-native status in our sample than in the study population, possibly due to a language barrier, as understanding Spanish was necessary to participate in the study.

## 5. Conclusions

In this study, exclusive breastfeeding rates were lower than the international recommendations. The factors associated with higher rates of EBF included being over 25 years old, having a university education, and having a partner with paternity leave. In contrast, mental illness, breast surgery, and maternal hypertension were associated with increased formula feeding. No relationship was found between the use of accessories and the cessation of or increased exclusive breastfeeding.

We propose training and updating professionals on predictive factors for BF difficulties and community educational interventions to increase awareness of the importance of breastfeeding as well as individualized counseling and educational interventions aimed at vulnerable populations (young mothers, those with lower education or pathologies, and those performing MF who have greater difficulty achieving EBF).

Both maternity and paternity leave have a positive impact on BF, but the situation is still far from optimal. Government and social policies that support and protect BF and work–life balance are essential.

## Figures and Tables

**Figure 1 nutrients-17-03109-f001:**
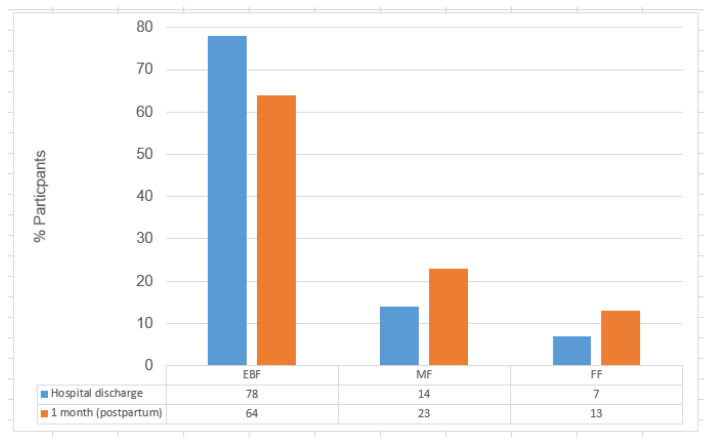
Transitions in the prevalence of infant feeding types (EBF, MF, and FF): hospital discharge vs. 1 month (n = 411).

**Table 1 nutrients-17-03109-t001:** Descriptive statistics of sociodemographic and clinical data.

Factors	N	Categories	n	%
Age (group)	407	≤25	34	8.35
		26–35	228	56.02
		≥36	145	35.63
Gender	409	M	406	99.27
		F	3	0.73
Education level	342	No studies or primary	74	21.64
		Compulsory secondary	29	8.48
		Vocational training	90	26.32
		University	149	43.57
Partner (id)	411	M	395	96.11
		F	2	0.49
		Non-binary	1	0.24
		Without partner	13	3.16
Place of birth (mother)	410	Spain	291	70.98
		Out of Spain	119	29.02
Religion (mother)	407	Christian	174	42.75
		Islam	22	5.41
		Buddhism	2	0.49
		Agnostic	142	34.89
		Other	67	16.46
Number of children	384	0	209	54.43
		1	128	33.33
		2	32	8.33
		≥3	15	3.91
Paid work (mother)	408	Yes	343	84.07
Maternity leave	408	Yes	342	83.82
Duration of maternity leave	338	<16 weeks	18	2.07
		≥16 weeks	331	97.93
Paternity (partner) leave	405	Yes	331	81.73
Duration of paternity leave	317	<16 weeks	32	10.10
		≥16 weeks	285	89.90
Maternal pathology	411	Type 1 diabetes	6	1.46
		Type 2 diabetes	8	1.95
		Hypertension	14	3.41
		Digestive disease	8	1.95
		Autoimmune disease	7	1.70
		Endocrine disease	27	6.57
		Mental illness	10	2.43
		Breast surgery	3	0.73
		Other	52	12.65
Neonatal pathology	411	Baby with low birth weight	23	5.6
		Premature	18	4.38
		Hyperbilirubinemia discharge	19	4.62
		Admitted to the NICU	24	5.84

**Table 2 nutrients-17-03109-t002:** Fluctuations in feeding type from hospital discharge to 1 month postpartum.

Discharge	1 Month	n	%	Change
EBF	EBF	247	59.4%	No change
EBF	MF	59	14.2%	Get worse
EBF	FF	17	4.1%	Get worse
MF	EBF	14	3.4%	Get better
MF	MF	37	8.9%	No change
MF	FF	7	1.7%	Get worse
FF	EBF	0	0.0%	
FF	MF	0	0.0%	
FF	FF	30	7.2%	No change

Note: EBF: exclusive breastfeeding, MF: mixed feeding, FF: formula feeding.

**Table 3 nutrients-17-03109-t003:** Number of babies by daily formula milk intakes and volume administered in mixed feeding at one month.

	Volume of Feeds (mL)									
N feeds per day	10	30	50	70	90	110	130	150	180	Total
1	1	1	4	3	8	1	1	0	0	19
2	1	3	3	2	3	1	0	1	0	14
3	0	1	1	2	4	2	1	1	1	13
4	0	5	1	1	3	1	0	0	0	11
5	0	1	0	1	2	0	1	1	0	6
6	0	0	3	2	2	2	3	0	0	12
7	0	0	0	0	2	0	0	1	0	3
8	0	3	1	2	3	0	3	0	0	12
9	0	1	0	0	3	0	0	0	0	4
Total	2	15	13	13	30	7	9	4	1	94

**Table 4 nutrients-17-03109-t004:** Number of babies by daily formula milk intakes and volume administered at one month of age in formula feeding.

	Volume of Feeds (mL)						
N feeds per day	50	90	110	130	150	180	TOTAL
5	0	0	0	1	0	0	1
6	0	3	1	0	1	1	6
7	0	4	5	2	**2**	1	14
8	0	8	6	4	**3**	0	21
9	1	4	1	5	**0**	1	12
TOTAL	1	19	13	12	6	3	54

**Table 5 nutrients-17-03109-t005:** Preventive factors and barriers to breastfeeding and mixed feeding in the first month of life.

Factor		Group	n	Feeding Type			Differences	
				EBF	MF	FF	chi2	*p*-Value
				Prevalence (%)				
**Age (group)** (N = 407)		≤25	34	44.12	23.53	32.35	**12.41**	**0.015**
		26–35	228	64.91	23.25	11.84		
		≥36	145	65.52	23.45	11.03		
**Education level** (N = 342)		No education or primary	74	54.05	32.43	13.51	**15.21**	**0.019**
		Compulsory secondary	29	62.07	20.69	17.24		
		Vocational training	90	51.11	28.89	20.00		
		university	149	72.48	16.78	10.74		
**Place of birth** (N = 410)		Spain	291	62.89	20.27	16.84	**13.96**	**0.001**
		Outside of Spain	119	65.55	30.25	4.20		
**Paternity (partner) leave** (N = 405)		No	75	54.05	24.32	21.62	**6.06**	**0.048**
		Yes	336	65.86	22.66	11.48		
**Maternal pathology** (N = 411)	Diabetes (combined) *	No	397	64.48	22.17	13.35	5.92	0.052
		Yes	14	42.86	50.00	7.14		
	**Hypertension**	No	397	64.23	23.43	12.34	**6.54**	**0.038**
		Yes	14	50.00	14.29	35.71		
	Autoimmune disease	No	404	64.11	22.52	13.37	4.99	0.082
		Yes	7	42.86	57.14	0.00		
	**Mental illness**	No	401	64.34	23.69	11.97	**20.41**	**<0.001**
		Yes	10	40.00	0.00	60.00		
	**Breast surgery**	No	408	63.97	23.28	12.75	**7.72**	**0.021**
		Yes	3	33.33	0.00	66.67		
	Other	No	232	64.62	23.68	11.70	5.21	0.074
		Yes	30	57.69	19.23	23.08		

Note: * Both types of diabetes combined here. Only significant (**in bold**) or marginal (*p* < 0.10) factors of the study (Table 1) are presented; the two types of diabetes in this study are combined here. Chi-squared values and corresponding *p*-values refer to Wilks’ test assessing differences in infant feeding prevalences.

## Data Availability

The data presented in this study are available on request from the corresponding author.

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
