# Peer review of "Characteristics, Preventive Factors, and Barriers to Breastfeeding and Mixed Feeding in the First Month of Life in Barcelona: The Multicenter Observational Study GREEN MOTHER"

_nutrients, 2025, doi:10.3390/nu17193109_

Round 1
Reviewer 1 Report
Comments and Suggestions for Authors
Dear authors,
The article entitled "Characteristics, preventive factors, and barriers to breastfeeding and mixed feeding in the first month of life in Barcelona: the multicenter observational study GREEN MOTHER" presents the results of a multicenter observational study, GREEN MOTHER, conducted in Barcelona.
The INTRODUCTION chapter strongly motivates the need for this study, using current references and rigorously explaining the international context.
In my opinion, this chapter is long and presents a lot of statistical data that could be summarized. There are also some repetitions (e.g., the advantages of breastfeeding).
The METHODS chapter contains a rigorous presentation of the methodology and a correct statistical analysis. I suggest that the description of the questionnaires be more concise. In addition, the loss of participants during follow-up could be analyzed.
The RESULTS chapter presents clear, well-organized data, with protective and risk factors clearly highlighted.
The tables with milk volumes are too detailed and difficult to read. Mixed feeding should be better analyzed.
Line 258 contains < PLEASE INSERT TABLE 4 HERE>.
Those lost to follow-up are not analyzed.
The DISCUSSION chapter is well contextualized internationally and correctly identifies the social and health implications. The chapter is long and contains repetitive information. I suggest using a table to summarize this repetitive information.
The LIMITATIONS subchapter highlights some of the study's shortcomings.
The CONCLUSIONS highlight how the objectives were achieved and make proposals for public policy.
The REFERENCES are numerous and up to date, covering both international and Spanish studies.
Good luck!
Author Response
We appreciate the reviewer suggestions and we attach our point-to-point replies.

Reviewer 2 Report
Comments and Suggestions for Authors
Dear Authors
After studying the article, we have some suggesting that could help in publishing.
Please read carefully our comments.
Best regards
Review
Characteristics, preventive factors and barriers to breastfeeding and mixed feeding in the first month of life in Barcelona
Dear Authors
After studying the article, we have some suggesting that could help in publishing.
In the Abstract
Prefer to write: Non-native (foreign-born) status and not terms (e.g., “foreigners”) could be more precise or culturally sensitive.
Some repetition or wordiness could be trimmed:
Prefer to write: 79% of women were exclusively breastfeeding, 14% practiced mixed feeding, and 7% used formula instead of: 79% of women were exclusively 40 breastfeeding, 14% were mixed breastfeeding, and 7% were formula feeding at hospital 41 discharge.
Strengths of abstract: Well-structured and logically sequenced.
- Incorporates robust citations and data.
- Clearly outlines the public health importance of breastfeeding.
- Ends with a clear and focused study aim.
Introduction
Lines 54–60: Recommendations
Suggestions:
- Line 55: The full name of UNICEF is not necessary after the acronym is widely known. Consider changing: “UNICEF”
- Line 57-58: Sentence is a bit long and awkward. Consider rephrasing:
"...continued breastfeeding (BF) with the gradual introduction of complementary foods, as appropriate, for as long as mother and child wish to continue." ➤ "...with the gradual introduction of complementary foods, continuing as long as mutually desired by mother and child."
Lines 61–70: Benefits of Breastfeeding
"BF provides enormous benefits..."
- Line 63: Avoid "better IQ" – too vague and potentially misleading. Consider: "...and possible cognitive advantages."
- Line 65: "Helps with birth spacing» not formal. Consider: "...and contributes to natural birth spacing."
- Line 70: “The superiority of BF versus FF is evident.” It is biased wording for a scientific paper. Consider: “The benefits of BF compared to FF are well documented.”
Lines 71–75:
"The UNICEF report... shows that 95% of babies are breastfed..."
- Line 72: Consider clarity: “just 4% were never breastfed” → Rephrase: "Only 4% of babies were never breastfed in low- and middle-income countries..."
- Line 74: The contrast between high-income and low-income countries is important — could be emphasized more clearly.
Lines 76–84: National Data (Spain & Catalonia)
"In Spain, less than 80% of babies started BF..."
- Line 76: "Started BF" → Use: "initiated breastfeeding"
- Line 82–83: Add clarity. Consider: "...94.8% of mothers were breastfeeding, of whom 75% were exclusively breastfeeding."
Lines 85–95: Mixed Feeding "MF involves combining or complementing BF with FF..."
- Line 85: Good explanation. Consider adding: “often used during the transition period or in response to challenges with exclusive breastfeeding.”
- Line 92–94: Define "Total Breastfeeding" earlier or in a separate sentence for clarity.
Lines 96–100: Reasons for Mixed Feeding "A recent systematic review identified..."
Very informative paragraph. Well organized.
- Line 99: “Returning to work or advice from healthcare professionals” – consider breaking this into two examples for clarity. Suggest: “...returning to work or receiving advice from healthcare professionals.”
Lines 101–109: Australian and Italian Studies "In Australia, R. Reynolds reported..."
- Line 103: Avoid listing characteristics without connecting them. Consider: "Women with lower education levels, higher BMI, and smoking habits were less likely to continue EBF."
- Line 108: “Already reporting such difficulties within the first month” → Use: "...reported these issues within the first month postpartum."
Lines 110–117: U.S. and U.K. Data "Odom also described..."
- Line 113: “Concluded that 60% of mothers...” → Rephrase: "...found that 60% of mothers stopped breastfeeding earlier than they had intended."
- Line 117: Better flow by combining similar reasons: "...25% cited discomfort and maternal fatigue, and 24% cited insufficient milk supply as reasons for early cessation."
Line 122: " to identify BF rates at hospital discharge and one month postpartum and analyze fluctuations in BF during this period and any associated factors in the region of Catalonia. – Suggest: "...analyze changes in exclusive, mixed, and formula feeding between hospital discharge and one month postpartum..."
Suggestions:
- Terminology
- Stick to either "BF / EBF / MF / FF" or full terms (e.g., “breastfeeding, exclusive breastfeeding...”) — ensure consistency across sections.
- 2. Cultural Sensitivity
- Consider replacing “foreigners” (from the abstract) with “foreign-born women” or “women of non-native origin” in the full text as well, if repeated.
2.1 Design and Study Setting (Lines 127–133)
"This was a multicenter, cross-sectional study with consecutive sampling of pregnant women in their third trimester and postpartum mothers at their first postpartum checkup..."
The phrase “cross-sectional” may not be accurate since you're collecting data at multiple time points (baseline, 7–10 days, 4–6 weeks).
“The cross-sectional study has an identical structure to the cohort study except that the exposures and outcomes are measured at the same time (i.e. cross-sectionally), whereas in a cohort study outcomes are typically measured after the exposure/s has been measured (i.e. longitudinally)”. Mann CJ. Observational research methods. Research design II: cohort, cross sectional, and case-control studies. Emerg Med J. 2003 Jan;20(1):54-60. doi: 10.1136/emj.20.1.54.
Consider calling it a “prospective observational cohort study” instead of "cross-sectional" for precision.
2.3 Inclusion and Exclusion Criteria (Lines 149–157)
Very commendable inclusion of illiterate but verbal participants — ethically sound and inclusive.
Be explicit that informed consent was verbal or assisted when necessary.
Suggested: illiterate women who could communicate verbally in these languages were included, with assistance from the principal investigator in completing the questionnaires.
2.5 Measurements (Lines 167–188)
"Three structured questionnaires were designed to collect data on..."
Descriptions of instruments and tools are appropriate, but lists could be formatted more cleanly. Suggestions: Add a brief note about validation of the questionnaires if applicable.
2.5 Data Analysis (Lines 189–198)
"A univariate analysis was used for descriptive statistics..."
Suggestions: Specify if normality test (e.g., Shapiro-Wilk) was performed for continuous variables.
Consider rephrasing “dynamic changes” for better clarity.
3.1. Sociodemographic and Clinical Description of the Participants
Table 1 is detailed and helpful.
Table Formatting: The structure of Table 1 is a bit chaotic. Variable labels, categories, and values should be consistently aligned.
Break down large tables if needed (e.g., separate maternal and neonatal factors).
3.2. Infant Feeding Results
- Clarity in Describing Trends:
Phrases like “fluctuations or dynamic changes” are vague. Use precise terms like "transitions" or "feeding trajectory changes."
"Figure 1 shows a decline in exclusive breastfeeding (EBF) from 79% at hospital discharge to 64% at one month postpartum (χ²=281.8, p<0.001), while mixed feeding (MF) increased from 14% to 23%, and formula feeding (FF) rose from 7% to 13%."
Line 233: Figure 2? You mean Figure 1?
- Interpret Results, Don’t Just List Them:
Mention that EBF is decreasing, which may be due to early challenges or social factors.
Suggestion: "These findings indicate a significant early postpartum decline in exclusive breastfeeding, with a noticeable shift toward mixed and formula feeding in the first month. This trend may reflect early breastfeeding difficulties or return-to-work pressures."
Recommendations:
- Ensure Figure 1 and Table 2 are clearly labelled and placed in appropriate locations in the text.
- Consider adding a brief interpretation at the end of the Results section to bridge into the Discussion.
It is confusing table 3 and table 4. Try to have clarity in description of the table 3 and 4.
Line 258. Erase the : < PLEASE. INSERT TABLE 4 ABOUTHERE>
3.3 Bivariate Associations Between Sociodemographic/Clinical Factors and Feeding Type
Solid analysis of categorical variables (age, education, birthplace, etc.). You’ve identified important predictors of EBF or FF.
Tables need better formatting:
- Place variable names as headers and align percentages better.
- Provide clear footnotes for abbreviations and what chi² and p-values refer to.
- Interpret key findings:
- Younger mothers (<25) have far lower EBF and higher FF—explain this trend.
- Highlight which clinical factors most strongly impacted feeding.
3.4 Multivariate Analysis
- Clear reporting of Relative Risk Ratios (RRRs) with 95% CIs.
- Appropriate baseline comparison (FF as reference group).
- Statistical clarity: (Line 303). Double-check for repeated or mistaken CI values (e.g., RRR = 1.8; 95% CI: 3.5–58.4 is likely a typo).
- Interpret each result:
Don't just state "association" — explain the direction and practical implication.
Discussion:
- EBF decreased significantly in the first month postpartum (79% → 64%).
- Most feeding changes were from EBF to MF.
- Breast pump use was associated with feeding stability.
- Factors positively associated with EBF:
- Maternal age ≥26
- University education
- Being born outside Spain
- Paternity leave
- Barriers to EBF:
- Maternal mental illness
- Breast surgery
- Hypertension
Introductory Paragraph (Lines 313–319)
This article analyzes the prevalence of different types of infants feeding and the fluctuations between them from birth to one month of age...
Replace terms like "this article" → use “this study” or “our study” to keep an academic tone.
Add context (lines 317-318): briefly compare your feeding rates with WHO or national targets.
- (Lines 320–324)
Inconsistent logic: You state that young women breastfeed more (Ferreira), but also that they wean early.
Make point clear: Young maternal age is a risk factor for early weaning.
- Education Level (Lines 325–329)
Be more assertive in your interpretation (e.g., higher education → greater health literacy). Connect education level to empowerment and informed decision-making.
- Partner Support & Leave (Lines 344–361)
- Unclear reasoning on why maternity leave didn't show significance.
- Some sentences repeat ideas.
- 5. Health Conditions (Lines 362–369)
- Need to separate mental illness vs. physical pathology.
- Prematurity and Birth Weight (Lines 370–373)
- The logic is weak; state your sample lacked enough variation or power.
- 7. Feeding Fluctuations (Lines 374–384)
- Sentence structure is long and hard to follow.
- 8. Breastfeeding Accessories (Lines 385–401)
- Better to split into pros and cons of accessories.
- 9. Formula Supplementation (Lines 402–410)
- Clarify motivation for MF: weight gain, flexibility, etc.
- Consider specifying the clinical implications (e.g., monitoring, counseling).
Author Response

(The authors gave the same response as above.)
